# How can deep transformers represent hierarchical languages? An expressivity analysis via bounded-depth grammars.

## Abstract

Deep neural networks are widely believed to derive their expressive power from their ability to form **hierarchical representations**, capturing progressively more abstract and compositional features across layers. In language modeling, **transformers** have emerged as the dominant architecture, with early layers capturing local syntactic patterns and later layers encoding more complex clause-level dependencies. While this intuition has shaped model design, there remains a lack of rigorous theoretical work demonstrating **how** deep transformers represent such hierarchical structures. In this work, we analyze the expressiveness of deep transformer models through the formal lens of bounded-depth, non-recursive context-free grammars. For this class of grammars, we explicitly construct transformers with positional attention whose depth grows linearly with grammar depth, while the neuron count scales with the number of derivation-tree shapes and quadratically with the number of production rules. Our theoretical results support the linear representation hypothesis by demonstrating that these architectures possess the structural capacity to encode abstract grammatical states into low-dimensional, linearly separable subspaces within the residual stream.

## 1 Introduction

Over the past two decades, breakthroughs in deep learning have revolutionized natural language processing tasks, such as language modeling (Brown et al. (2020); Devlin et al. (2019); Hoffmann et al. (2022)), question-answering (Lan et al. (2020); Beltagy et al. (2020); Zaheer et al. (2020)), and machine translation (Vaswani et al. (2017); Costa-Jussà et al. (2022); Fan et al. (2020)). While rule-based methods were widely used for next-word prediction before deep learning, neural language models have since consistently demonstrated superior performance on complex tasks (Bengio et al. (2003)). Transformers, characterized by their self-attention mechanisms, further advanced the field by enabling models to process entire sequences in parallel and dynamically focus on distant dependencies — capabilities that earlier recurrent or convolutional approaches struggled to achieve (Devlin et al. (2019); Brown et al. (2020)). Although larger models and novel architectures continue to yield empirical gains (Raffel et al. (2020); Hoffmann et al. (2022)), we still lack a firm theoretical understanding of *how transformers capture the complex hierarchical structures of language so effectively.*

Natural language exhibits hierarchical structure, which can be represented by context-free grammars: sentences are composed of phrases, which are composed of sub-phrases or words (Chomsky (1957)). While recursion allows the space of valid sentences to grow infinitely, human cognitive limits typically restrict parsing to bounded depths. Hierarchical representations exploit this structure, allowing models to decompose sequences into smaller subsequences, dramatically reducing the number of parameters required (Poggio et al. (2017)). Although prior work has shown that deep networks can leverage such compositional structures (Mossel (2016)), they have largely focused on restricted settings such as Dyck languages (Hahn (2020); Yao et al. (2021)) and finite state automata (Liu et al. (2023)), which lack the complex branching structures of real-world grammars.

Recent empirical studies show how transformers can internalize complex syntactic and semantic structures, offering strong support for the *linear representation hypothesis*. For instance, Saglam et al. (2025) demonstrates that large language models encode high-level semantic domains into low-dimensional, linearly separable subspaces, with this separability becoming distinctly more pronounced in the deeper layers of the network. In parallel, Allen-Zhu & Li (2025) and Zhao et al. (2023) show that transformers trained on synthetic context-free grammars (CFGs) can recover latent non-terminal symbols through linear probes, further indicating that abstract hierarchical rules are encoded as linear directions in the residual stream. While these works empirically validate the linear representation hypothesis, they do not theoretically explain why transformer architectures possess the structural capacity to explicitly represent these complex grammatical structures within low-dimensional linear subspaces.

**Contributions.** To address these limitations, we analyze transformer language models through the formal framework of length-uniform, non-recursive context-free grammars (CFGs) with a fixed depth, a setting that makes hierarchical relationships explicit (Strobl et al. (2024); Ackerman & Cybenko (2020)). For this class of CFGs, we provide an *explicit* construction of a transformer architecture with a positional attention mechanism whose depth grows linearly with grammar depth, while the neuron count scales linearly with $c$, the number of valid derivation-tree shapes, and quadratically with the number of production rules. We note that our simplified framework can be used to analyze hierarchical compositionality, as even with a restricted number of tree shapes, the underlying grammar can generate an exponentially large number of unique sentences.

Our framework provides a rigorous existence proof compatible with the "linear representation hypothesis" in the context of hierarchical language modelling. We formally prove that the architecture has the capacity to encode complex grammatical structures in low-dimensional spaces, with any non-terminal symbol at a given depth mapped to a vector in a subspace of the corresponding residual stream. While empirical models achieve this through dense superposition (Garg et al. (2026); Saglam et al. (2025)), our construction utilizes a simplified model with an orthogonal basis space. We construct transformers that naturally implement the bottom-up dynamic programming algorithm for parsing CFGs described in Allen-Zhu & Li (2025). Our proof relies on hard-coded attention heads that aggregate local syntactic contexts, mirroring the tree-building attention patterns empirically observed in trained models (Allen-Zhu & Li (2025)).

**Organization of the paper.** Section 2 summarizes related work on hierarchical modeling and formal language theory. Section 3 defines the class of CFGs we analyze (with examples) and formally states the problem setup for transformer models. In Sections 4 and 5, we present our main expressiveness results, showing that deep transformers can solve next-word prediction for context-free grammars with a fixed depth, and outline a constructive proof describing the function of attention heads and feedforward layers. In Section 6, we discuss limitations and further directions. The Appendix provides detailed proofs of our theoretical results.

## 2 Related work

**Hierarchical representations via deep learning.** A growing body of work has investigated the role of hierarchical representations in deep learning models. It is well understood that deep networks can compactly represent hierarchical functions using exponentially fewer parameters than shallow networks (Poggio et al. (2017); Zhao et al. (2017)). In certain settings, generative models grounded in hierarchical structure have been shown to be learnable via clustering-based techniques(Mossel (2016); Malach & Shalev-Shwartz (2018; 2020)). More recent work has demonstrated that deep networks trained via gradient descent can uncover these latent hierarchical structures implicitly (Allen-Zhu & Li (2025); Tomasini & Wyart (2024); Garnier-Brun et al. (2025)). We study how transformer models represent hierarchical structures through the mathematical framework of CFGs, a formalism well suited for modeling natural language.

**Modelling formal languages using transformers.** Several recent works have explored the computational capabilities of transformer models from the perspective of formal language theory. It has been shown in Perez et al. (2021) that encoder-decoder transformers can recognize all languages in the class P, i.e., those decidable by deterministic Turing machines in polynomial time. This result, which has been refined in Bhattamishra et al. (2020b); Merrill & Sabharwal (2024) and Sarrof et al. (2024); Yang et al. (2024), provides one of

the few known characterizations of which formal languages can be modelled by transformer architectures. Recent work has also investigated the ability of attention-based models to recognize formal languages such as $\text{Dyck}_k$, the language of well-balanced brackets with $k$ types (Hahn (2020); Yao et al. (2021); Bhattamishra et al. (2020a)), and whether transformers can recognize formal languages in the complexity class $\mathsf{AC}^0$ (Hao et al. (2022); Barcelo et al. (2024)), which consists of languages recognizable by families of Boolean circuits of constant depth and polynomial size. The present work is most closely related to Zhao et al. (2023), which shows that transformers can implement the inside-outside parsing algorithm for CFGs, and Liu et al. (2023), which analyzes the depth expressiveness of transformers through the lens of finite-state automata. Building on these results, we establish a direct connection between the depth of the transformer and the depth of the underlying grammar, and obtain novel upper bounds on the size of a transformer required to represent certain classes of CFGs.

## 3 Problem set-up

### 3.1 Preliminaries on language modelling

We begin by formalizing the language modelling task following Sarrof et al. (2024).

**Definition 3.1.** The *vocabulary* $\Sigma$ is a finite set consisting of all words. Let $\Sigma^*$ denote the set of all finite sequences consisting of words from $\Sigma$. Given a vocabulary $\Sigma$, a *language* $\mathcal{L}$ is a subset of $\Sigma^*$; its elements are called sentences. ∎

**Definition 3.2.** Given a vocabulary $\Sigma$, a *language model* is a function $f$ that takes in an arbitrary sequence $\underline{w} \in \Sigma^*$, and outputs a vector of probabilities $f(\underline{w}) = (\hat{f}(w|\underline{w}))_{w \in \Sigma}$. Here $\hat{f}(w|\underline{w})$ denotes the probability that the model $f$ will output the word $w$ given the input sequence $\underline{w}$. ∎

Many theoretical papers study the recognition problem (Bhattamishra et al. (2020a)), where the language model must determine whether or not a complete string belongs to a given language. However, recent empirical work in language modelling has focused on next-word prediction. We focus on the predictive modelling problem, and define what it means for $f$ to model a given language $\mathcal{L}$ (note that the recognition problem can be obtained as a special case of predictive modelling; see Section 3.2 of Sarrof et al. (2024)).

**Definition 3.3.** Given a vocabulary $\Sigma$ and a language $\mathcal{L} \subseteq \Sigma^*$, the set of *valid prefixes* of $\mathcal{L}$, denoted $\text{Prefix}(\mathcal{L})$, is defined as follows.

$$\text{Prefix}(\mathcal{L}) := \{\underline{w} \in \Sigma^* : \exists \underline{w}' \in \Sigma^*, \ \underline{w} \cdot \underline{w}' \in \mathcal{L}\}$$

For a given valid prefix $\underline{w} \in \text{Prefix}(\mathcal{L})$, we define $S(\underline{w})$, the set of all words in $\Sigma$ that can follow the sequence $\underline{w}$ in a sentence from $\mathcal{L}$, as follows.

$$S(\underline{w}) := \{w \in \Sigma : \underline{w} \cdot w \in \text{Prefix}(\mathcal{L})\} \qquad ∎$$

**Definition 3.4** (Predictive modelling). Let $\mathcal{L} \subseteq \Sigma^*$ be a language, and let $f$ be a language model as in Definition 3.2. For $\epsilon \geq 0$, we say that $f$ *predictively models* $\mathcal{L}$ with error at most $\epsilon$ if, for every valid prefix $\underline{w} \in \text{Prefix}(\mathcal{L})$,

$$\sum_{w \notin S(\underline{w})} \hat{f}(w \mid \underline{w}) \ \leq \ \epsilon \qquad ∎$$

### 3.2 Context-free grammars

Context-free grammars (CFGs) are a foundational class of formal grammars used to model the syntax of natural language. A CFG consists of a finite set of production rules that describe how strings in a language can be generated from a designated start symbol. Each rule specifies how a single non-terminal symbol can be replaced by a sequence of terminal and/or non-terminal symbols, allowing for recursive and nested derivations. Unlike regular grammars, CFGs can represent hierarchical and nested structures present in annotated corpora such as Penn Treebank (Charniak (1996)), where the production rules can be obtained from the parse trees in the corpus (see also Nederhof & Vogler (2012); Koller et al. (2008)). Below, we formally define a CFG following Allen-Zhu & Li (2025).

| Component | Symbols / Rules |
|---|---|
| **Non-terminals (Level 3)** | $N_3 = \{a_1^3,\, a_2^3,\, a_3^3,\, a_4^3,\, a_5^3\}$ |
| **Non-terminals (Level 2)** | $N_2 = \{a_1^2,\, a_2^2,\, a_3^2,\, a_4^2,\, a_5^2, a_6^2\}$ |
| **Terminals** | $N_1 = \Sigma = \{a_1, a_2, a_3, a_4, a_5, a_6, a_7, a_8, a_9, a_{10}\}$ |
| **Production Rules** | $S \;\rightarrow\; a_1^3\, a_4^3\, a_5^3 \;\;\mid\;\; a_2^3\, a_3^3\, a_4^3$ |
| | $a_1^3 \;\rightarrow\; a_1^2\, a_6^2; \qquad a_2^3 \;\rightarrow\; a_3^2\, a_4^2$ |
| | $a_3^3 \;\rightarrow\; a_5^2\, a_1^2\, a_2^2; \qquad a_4^3 \;\rightarrow\; a_3^2\, a_4^2 \;\mid\; a_4^2\, a_5^2$ |
| | $a_5^3 \;\rightarrow\; a_1^2\, a_2^2$ |
| | $a_1^2 \;\rightarrow\; a_1\, a_6 \;\mid\; a_5\, a_3 \;\mid\; a_2\, a_3; \qquad a_2^2 \;\rightarrow\; a_{10}\, a_8\, a_9 \;\mid\; a_1\, a_7\, a_9$ |
| | $a_3^2 \;\rightarrow\; a_5\, a_2 \;\mid\; a_6\, a_8; \qquad a_4^2 \;\rightarrow\; a_7\, a_4 \;\mid\; a_3\, a_9 \;\mid\; a_7\, a_8$ |
| | $a_5^2 \;\rightarrow\; a_3\, a_4 \;\mid\; a_{10}\, a_4 \;\mid\; a_3\, a_{10}; \qquad a_6^2 \;\rightarrow\; a_5\, a_8 \;\mid\; a_6\, a_5$ |

Table 1: A Depth-3 CFG that can be used to generate the two derivation trees below.

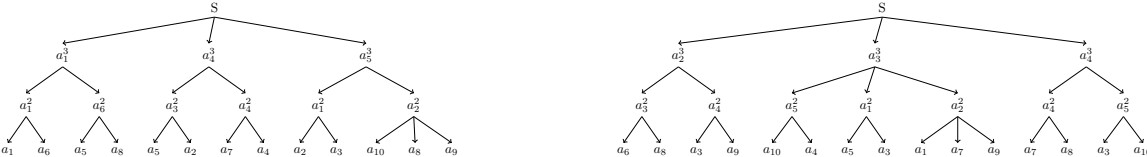

Figure 1: Two derivation trees with different shapes. All nodes have degree 2 or 3.

**Definition 3.5.** A *context-free grammar* $\mathcal{G}$ consists of the following components $(\Sigma, N, P, S)$:

- $\Sigma$ is a finite set of terminal symbols (the alphabet of the language).

- $N$ is a finite set of non-terminal symbols, which is disjoint from $\Sigma$. One distinguished symbol $S \in N$ is called the *start symbol*.

- $P$ is a finite set of production rules. Each rule is of the form $A \rightarrow \beta$, where $A \in N$ is a single non-terminal symbol, and $\beta$ is composed of symbols from $N \cup \Sigma$.

Given a context-free grammar $\mathcal{G}$, the language $\mathcal{L}(\mathcal{G})$ generated by the grammar is defined as follows:

- A *derivation tree* is a hierarchical structure obtained by recursively applying production rules that start with the root node labeled with the start symbol $S$. Internal nodes are labeled by non-terminal symbols and are expanded according to rules in $P$, and all leaf nodes correspond to terminal symbols.

- The set $\mathcal{L}(\mathcal{G})$ consists of all sequences of terminal symbols obtained by reading the leaves of a derivation tree from left to right. ∎

Following Section 2 of Allen-Zhu & Li (2025), we consider CFGs with a fixed depth $d$, where every derivation tree generated by the grammar has height exactly $d$. This constraint naturally induces a hierarchical structure with $d$ levels of composition, making such grammars a useful abstraction for analyzing how deep language models, such as transformers, can represent compositional structure across multiple layers. This framework can also be extended to grammars where root-to-leaf paths have length at most $d$, by inserting duplicate copies of a non-terminal along shorter branches so that every path is extended to length exactly $d$.

**Definition 3.6.** We say that the CFG $\mathcal{G}$ has *depth $d$* if every derivation tree generated by $\mathcal{G}$ satisfies the following property: for every root-to-leaf path, the path length (i.e., the number of edges from the root node to a leaf node) is exactly $d$. Let $N_i$ denote the set of symbols that appear at level $i$ of any derivation tree, starting from the bottom (in particular, let $N_1 = \Sigma$ be the set of terminal symbols, and $N_{d+1} = \{S\}$ consist of the start symbol).

**Definition 3.7.** Let $\mathcal{G} = (\Sigma, N, P, S)$ be a CFG. If $\mathcal{L}(\mathcal{G}) \subseteq \Sigma^*$ is the generated language, let

$$L = \max\{ |w| : w \in \mathcal{L}(\mathcal{G})\},$$

be the *maximal sentence length*. For each non-terminal $A \in N$, define its *degree*

$$\deg(A) = \max_{(A \to \beta) \in P} |\beta|,$$

where $|\beta|$ is the number of symbols on the right-hand side of the production. We say that $\mathcal{G}$ is *uniform* if for every non-terminal $A \in N$, all of its production rules have the same length $\deg(A)$. We say that $\mathcal{G}$ has *maximal rule count $M$* if the following holds: for each depth $i$ ($2 \leq i \leq d$), there are at most $M$ distinct production rules where the non-terminal symbol on the left is in $N_i$. We define the *maximal branching factor $B$* as the largest number of symbols that can appear on the right-hand side of any production rule.

**Definition 3.8.** Let $\mathcal{G}$ be a CFG, and let $T$ be a derivation tree for $\mathcal{G}$. The *shape* of $T$, denoted $\text{shape}(T)$, is the ordered, unlabeled rooted tree obtained by forgetting all non-terminal symbol labels in $T$ while retaining the parent–child relations and the left-to-right order of siblings. Two derivation trees $T_1$ and $T_2$ are said to have the same shape precisely when there exists an isomorphism of ordered unlabeled trees between $\text{shape}(T_1)$ and $\text{shape}(T_2)$. We denote by $\mathcal{C}(\mathcal{G})$ the set of all possible shapes of derivation trees in $\mathcal{G}$.

While our main theorem explicitly assumes derivation trees of a fixed depth $d$, this framework naturally extends to context-free grammars with a bounded depth less than or equal to $d$. This is done by augmenting the original grammar with duplicate non-terminal symbols to pad any prematurely terminating derivation branches. Specifically, if a valid derivation path yields a sequence at a shallower depth, we can introduce a chain of duplicate non-terminals converting that derivation tree into an equivalent fixed-depth tree.

It is important to note that our theoretical framework analyzes context-free grammars with bounded depth, and excludes recursive CFGs. A recursive CFG allows for unbounded nesting, where a non-terminal symbol can be expanded to a sequence containing itself (e.g., $A \Rightarrow^* \alpha A \beta$), thereby generating derivation trees of unbounded depth. However, natural language rarely exhibits deep recursive nesting due to limitations on human working memory, and our bounded-depth assumption is a realistic constraint for modeling natural language.

### 3.3 Transformer models

We assume familiarity with the transformer architecture introduced in Vaswani et al. (2017), and definitions are presented below for clarity. A *transformer block* first applies multi-head self-attention, followed by a position-wise feedforward neural network, wrapping each sub-block in residual connections. A *transformer model* is obtained by composing transformer blocks and optionally appending a feedforward network with one or more fully connected layers before the output head, motivated by recent work in Press et al. (2020) showing that reordering and stacking multiple feedforward layers can enhance the representational capacity of transformers. We also use a relative positional attention mechanism defined below using the random synthesizer construction in Section 3.1 of Tay et al. (2020), where the attention weights depend only on the position of the tokens and not their content (see also Shaw et al. (2018)).

**Definition 3.9** (Multi-Head Attention with Relative Positional Weights)**.** Let $X \in \mathbb{R}^{n \times d}$ be the input sequence consisting of $n$ vectors in $\mathbb{R}^d$. Fix a maximum sequence length $L_{\max} \geq n$. For each head $h = 1, \ldots, H$, we learn

$$R_h \in \mathbb{R}^{L_{\max} \times L_{\max}}, \qquad W_h^{(V)} \in \mathbb{R}^{d \times d_V},$$

where the relative positional matrix $R_h$ is lower triangular and assigns a scalar weight to each ordered pair of positions in $\{1, \ldots, L_{\max}\}$. Letting $R_h^{[1:n,1:n]}$ be the sliced positional matrix, we compute the value matrix $V_h = X W_h^{(V)} \in \mathbb{R}^{n \times d_V}$ and the head output

$$\text{head}_h = R_h^{[1:n,1:n]} V_h \in \mathbb{R}^{n \times d_V}.$$

The multi-head output is

$$\text{MHA}(X) = \text{Concat}(\text{head}_1, \ldots, \text{head}_H) W^{(O)},$$

where $W^{(O)} \in \mathbb{R}^{Hd_V \times d}$ is an output projection. ∎

**Definition 3.10.** A position-wise feedforward network (FFN) with input/output width $d$ and hidden width $d_{\text{ff}}$ is the function

$$\text{FF} : \mathbb{R}^d \to \mathbb{R}^d, \qquad x \mapsto W^{(2)}\sigma(W^{(1)}x + b^{(1)}) + b^{(2)},$$

with $W^{(1)} \in \mathbb{R}^{d_{\text{ff}} \times d}$, $b^{(1)} \in \mathbb{R}^{d_{\text{ff}}}$, $W^{(2)} \in \mathbb{R}^{d \times d_{\text{ff}}}$, $b^{(2)} \in \mathbb{R}^d$, and $\sigma$ the ReLU applied elementwise.

**Definition 3.11.** A transformer layer with $H$ heads and width $d$ is a function defined as follows.

$$\text{Layer} : \mathbb{R}^{n \times d} \to \mathbb{R}^{n \times d}, \text{Layer}(A) = A''$$

$$A' = \text{MHA}(A) + A$$

$$A'' = \text{FF}(A') + A'$$

Here MHA is the multi-head attention mechanism, and FF is a position-wise feedforward network (FFN). The terms $+A$ and $+A'$ are residual connections. ∎

**Definition 3.12** (Transformer Model)**.** A *transformer model* with $L$ layers, $H$ attention heads per layer, and width $d$ is a function $f : \mathbb{R}^{n \times d} \to \mathbb{R}^k$ defined as the composition of $L$ transformer layers, a stack of one or more fully connected feedforward layers, and a linear output layer followed by a softmax. ∎

## 4 Theoretical results

In this section, we present the main results that bound the expressiveness of transformers by analyzing predictive modeling for languages generated from a class of uniform, non-recursive CFGs of fixed depth. In Section 4.1 we present a constructive result, building a transformer that predictively models a class of CFGs, using a simplified positional attention mechanism. In our construction, the number of neurons required grows linearly with the number of derivation tree shapes and quadratically with the number of production rules, while the model's depth grows linearly with the depth of the grammar. Our proof, based on the concept of hierarchical compositionality, illustrates how different layers of a transformer encode the hierarchical structure present in CFGs, with lower layers capturing simpler syntactic structures and upper layers capturing progressively more complex structures. In Section 4.2, we discuss connections with existing work comparing the expressiveness of transformers with language models based on $n$-grams and finite state automata. We refer the reader to Section 5 for an outline of the proofs, and the Appendix for detailed proofs.

### 4.1 Statement of main Theorem

We analyze a class of CFGs without recursive rules that generate finite languages, which can be modelled with a sufficiently large transformer using existing results on memorization (Zhang et al. (2017)). However, the number of neurons required by naive memorization grows exponentially. Our work instead establishes much sharper structural bounds on transformer size by leveraging transformers' ability to represent hierarchical structure. This differs from existing work analyzing whether transformers can model infinite languages obtained from CFGs with recursive rules, such as generalisations of Dyck languages (Yao et al. (2021); Hahn (2020)) and more general formal languages that can be recognized by a Turing machine (Perez et al. (2021); Bhattamishra et al. (2020b)). While our bounds on the model size depend on $c$, we note that even with a restricted set of tree shapes, the total number of valid sentences in the language can grow exponentially.

**Theorem 4.1.** *Let $\mathcal{G} = (\Sigma, N, P, S)$ be a uniform context-free grammar with depth $d$, maximal rule count $M$, maximal branching factor $B$, and let $c = |\mathcal{C}(\mathcal{G})|$. Given an $\epsilon > 0$, there exists a transformer language model $f_{\mathcal{T}} : \Sigma^* \to \mathbb{R}^{|\Sigma|}$ consisting of $d$ transformer blocks followed by $5d + 7$ feedforward layers satisfying the following properties.*

- *The transformer $f_{\mathcal{T}}$ predictively models the language $\mathcal{L}(\mathcal{G})$ with error at most $\epsilon$.*

- *The transformer block in each layer has hidden dimension $cd$, and consists of $c$ attention heads, followed by a feedforward neural network with $M$ neurons. Each attention head has value dimension 1, and the $W^{(O)}$ matrix has dimension $c \times cd$. For each attention head $h$, the relative positional matrix $R_h$ has at most $B$ non-zero entries in each row.*

- *Each of the $5d + 7$ feedforward layers has at most $4cBM^2$ neurons.* ∎

The key results of Zhao et al. (2023) show that transformers can implement the inside-outside algorithm for parsing CFGs, and the above theorem differs in two key aspects. Firstly, while Zhao et al. (2023) focuses on encoder-only transformers such as BERT (Devlin et al. (2019)), our results analyze decoder-only transformers which are more widely used in language modelling (Brown et al. (2020)). Secondly, while Zhao et al. (2023) establishes bounds on the number of parameters in a transformer needed to model CFGs, the number of layers in their construction grows linearly with the maximal sentence length $L$. In our construction, the number of layers in the transformer grows linearly with the depth of the CFG, which is an improvement since the maximal sentence length can grow exponentially with the depth of the grammar. We note that due to the dependency on $c$, our key result does not yield tight bounds for complex real-world corpora like the Penn Treebank (Marcus et al. (1993)), but formalizes the core mechanics of hierarchical compositionality for simplified grammars.

In order to prove a similar statement with standard attention blocks that don't use the simplified relative positional encoding, one can follow the approach from the proof of Theorem 4.1 in Svete & Cotterell (2024). This construction augments the input token representations with explicit positional embeddings, and the query and key projection matrices are designed such that the inner product depends solely on the position of the tokens. By applying a scaling factor, the softmax operation approximates a hard attention mechanism, allowing the attention heads to select tokens at the precise positions required to identify the relevant grammatical constituents. In this setting, the size of the attention heads would scale linearly with the context window length.

Our construction of the transformer model has the property that each input sequence is processed using a sparse circuit with a single attention head in each transformer block. This aligns with empirical work extracting circuits (Olah et al. (2020); Elhage et al. (2021)), which are sparse computational subgraphs within deep learning models that can perform specific tasks. Our construction of attention heads also has the property that each token only attends to a sparse subset of tokens, which aligns with existing work on the effectiveness of sparse attention mechanisms (Roy et al. (2020); Beltagy et al. (2020); Yuan et al. (2025)).

### 4.2 A comparison with $n$-grams and finite state automata.

We first compare the above results with Theorem 4.1 from Svete & Cotterell (2024), which constructs transformers with a single layer and a single head that represent $n$-gram language models (see also Nandakumar et al. (2025) for analogous results with state space models). While the languages in Theorem 4.1 can be modeled using $n$-gram rules, this typically requires an exponential number of rules. A depth-2 CFG resembles an $n$-gram model when it has depth 2 and width $(k, 1)$ with $k > n$. In this setting, the start symbol expands into exactly $k$ non-terminals, each of which expands to some number of terminal symbols. Our results can be viewed as an extension of Theorem 4.1 of Svete & Cotterell (2024) to deeper transformer models and deeper CFGs.

The class of CFGs that we analyze can be represented by finite state automata, and this representation plays a key role in the proof of our Theorem 4.1. The representational capabilities of transformers are explored in Liu et al. (2023) by analyzing their ability to represent finite state automata, with their key results showing that the size of the transformer required grows polynomially with the number of states in the finite state automata. However, when the CFGs in Theorem 4.1 are expressed as finite state automata, the number of states grows exponentially with the depth of the grammar, and our key results cannot be obtained from those of Liu et al. (2023). Specifically, converting a CFG with a set of non-terminals $N$ into an equivalent FSA up to a bounded derivation depth $d$ requires explicitly encoding all possible valid combinations of unresolved non-terminals into the automaton's state space $Q$. This flattening of the hierarchical derivation tree into a linear state machine yields a state space of size $|Q| = \mathcal{O}(|N|^d)$. This exponential blow-up highlights the inefficiency of modeling deep context-free grammars using finite state automata, and our construction bypasses this by natively leveraging the depth of the transformer to sequentially process grammatical reductions level-by-level.

### 4.3 Comparison with empirical work

Our results above on the expressiveness of transformers provide a theoretical framework for the controlled experiments conducted in Allen-Zhu & Li (2025), where the authors trained transformer models with the GPT2-small architecture, using 12 layers with dimension 768, from scratch on synthetic context-free grammars. To ensure the models learn true hierarchical representations rather than local heuristics, the authors designed deeply nested CFGs which generate sequences of up to 729 tokens in length, with over $10^{80}$ valid string combinations in total. The empirical finding that these decoder-only transformers can achieve near-perfect generation accuracy on these highly complex, ambiguous sequences shows that the model does not require exponential scaling to process recursive grammatical structures, as our explicit bounds predict.

In Allen-Zhu & Li (2025), the authors observe that transformers trained on synthetic data from CFGs naturally develop specialized attention heads that execute a bottom-up parsing strategy by dynamically attending to the boundaries of grammatical constituents and gathering information from the child nodes of the derivation tree to form the parent non-terminal symbol. This aligns with the explicit construction of self-attention in our proof of Theorem 4.1, where attention heads are explicitly constructed to selectively aggregate the local syntactic context of child nodes in order to resolve the parent non-terminals, layer by layer.

In particular, Section 5.3 of Allen-Zhu & Li (2025) describes an empirical dynamic programming algorithm that computes states denoted as $DP(k, j, a)$ to track whether a non-terminal symbol $a$ spans the sequence from token $k$ to $j$. Furthermore, in Section 4.1 they demonstrate that these intermediate $DP(k, j, a)$ symbols can be recovered from the residual stream in the final layers using simple linear probes. Our constructive proof formalizes these empirical findings by representing the states $DP(k, j, a)$ using neurons within the hidden layers of the transformer. We construct multi-head attention blocks and feedforward neural networks in each layer to implement the recursion used to compute $DP(k, j, a)$ symbols from lower-level constituents.

## 5 Proofs.

In this section, we outline the proof of the key result, Theorem 4.1.

Given an input sequence $\underline{w} = (w_j)_{1 \leq j \leq k}$, we denote by $z_i(\underline{w})$ its image after passing through the $i$-th transformer block. We decompose these vectors $z_i(\underline{w})$ in each transformer layer into $c$ blocks as follows: $z_i(\underline{w}) = \{z_{i,\underline{c}}(\underline{w})\}_{\underline{c} \in \mathcal{C}(\mathcal{G})}$. Here $z_{i,\underline{c}}(\underline{w}) \in \mathbb{R}^d$ for each $\underline{c} \in \mathcal{C}(\mathcal{G})$. The attention heads in the $i$-th transformer layer are indexed by a class $\underline{c} \in \mathcal{C}(\mathcal{G})$, and we denote by $a_i^c(\underline{w}) \in \mathbb{R}$ the image of $\underline{w}$ in the corresponding attention head within the $i$-th layer.

### 5.1 Segmenting input sequences using attention heads

For a fixed shape $\underline{c}$ of the derivation tree, begin with the input sequence $\underline{w}$ and repeatedly collapse every group of children that belongs to the same parent, working from the leaves upward. After this has been done and no further reductions are possible, define $E_{i,\underline{c}}(\underline{w})$ to be the sequence of non-terminal symbols at level $i$ appearing in the resulting string. The quantities $E'_{i,\underline{c}}(\underline{w})$ and $E^*_{i,\underline{c}}(\underline{w})$ are defined similarly (see Section 5.5 for an example, and the Appendix for a rigorous definition).

**Lemma 5.1.** *We can choose attention heads with the following property. Given an input sequence $\underline{w} = (w_1, \cdots, w_k)$ corresponding to the class $\underline{c}$, the output $a_i^{\underline{c}}(\underline{w})$ of the attention head indexed by $\underline{c}$ in the $i$-th layer depends only on $E'_{i,\underline{c}}(\underline{w})$ (i.e. if another input $\underline{w}'$ also corresponds to the class $\underline{c}$, then $a_i^{\underline{c}}(\underline{w}) = a_i^{\underline{c}}(\underline{w}')$ if and only if $E'_{i,\underline{c}}(\underline{w}) = E'_{i,\underline{c}}(\underline{w}')$).*

To prove the above Lemma, we choose the query–key projections for the attention head $a_i^c$ so that a token attends only to a group of tokens which are determined by the shape $\underline{c}$ of the derivation tree (see Section A.2 for a detailed proof). Because we omit the softmax in the attention mechanism and instead apply these deterministic, position-dependent masks, the attention score for this token and any token outside the specified group is zero. In order to prove a similar statement without using the simplified relative positional

encoding, one can follow the approach from the proof of Theorem 4.1 in Svete & Cotterell (2024), and use query-key matrices that encode positional information.

## 5.2 Encoding non-terminal symbols with feedforward networks

The next step is to encode non-terminal symbols by using the feedforward network component of the transformer block. To prove Lemma 5.2 below, we explicitly construct the feedforward layers to act as a discrete look-up table using memorization to encode the finite set of valid production rules in the context-free grammar. From Lemma 5.1, for a given derivation shape, the attention mechanism isolates and aggregates the necessary constituents, which are represented by the sequence $E'_{i,\underline{c}}(\underline{w})$. The feedforward network receives this aggregated output and matches it against its memorized rules, identifies the corresponding parent non-terminal symbol and encodes it in the $(i+1)$-st coordinate of the intermediate representation $z_{i,\underline{c}}(\underline{w})$. The transformer's residual skip connections ensure that the lower-level syntactic states in coordinates $j \leq i$ are preserved, and used in successive layers.

**Lemma 5.2.** *We can choose the weights of the transformer so that the following holds. Given an input sequence $\underline{w} = (w_1, \cdots, w_k)$ corresponding to the class $\underline{c}$, and for each $j \leq i$, the $j$-th coordinate of $z_{i,\underline{c}}(\underline{w})$ depends only on the string $E_{j,\underline{c}}(\underline{w})$. The $(i+1)$-st coordinate of $z_{i,\underline{c}}(\underline{w})$ depends only on the string $E^*_{i+1,\underline{c}}(\underline{w})$, and is 0 if $E^*_{i+1,\underline{c}}(\underline{w}) = \emptyset$. Further, the $j$-th coordinate of $z_{i,\underline{c}}(\underline{w})$ is 0 when $j > i+1$.*

Our proof of the above uses the below result from Zhang et al. (2017), which analyzes the memorization capacity of feedforward neural networks, and shows that they can fit an arbitrary finite set of input-output vectors (see also Yun et al. (2019)).

**Lemma 5.3.** *Let $\{(x_i, y_i)\}_{i=1}^K$ be a finite dataset where each input $x_i \in \mathbb{R}^n$ and each output $y_i \in \mathbb{R}^q$. Assume that all inputs are distinct, i.e., $x_i \neq x_j$ for $i \neq j$. Consider a feedforward neural network $f_\theta$ with one hidden layer of width $K$, ReLU activations, input dimension $n$, and output dimension $q$. Then there exists a choice of parameters $\theta$ such that $f_\theta(x_i) = y_i$ for all $i \in [K]$.*

## 5.3 Using stacked feedforward networks for next-word prediction

The final step is to use a position-wise feedforward network with $5d+7$ hidden layers to process the encodings of segments from the last transformer block and obtain vectors that encode the set of valid next words $S(\underline{w})$ for a given valid prefix $\underline{w} = (w_1, \cdots, w_k)$. Denote by $h_i(\underline{w})$ its image after passing through the $i$-th feedforward layer in the sequence; we decompose these vectors as follows: $h_i(\underline{w}) = (h_{i,\underline{c}}(\underline{w}))_{\underline{c} \in \mathcal{C}(\mathcal{G})}$.

Given an input sequence $\underline{w}$, consider the string of symbols obtained by concatenating $E_{d,\underline{c}}(\underline{w})$, $E_{d-1,\underline{c}}(\underline{w}), ..., E_{d-i+1,\underline{c}}(\underline{w})$. We define the set $N_{d-i+1,\underline{c}}(\underline{w})$ as the set of all non-terminals of $N_{d-i+1}$ which can be appended to this string to yield a sequence which can be expanded into a valid sentence. These sets can be computed inductively, which can be done using feedforward neural networks (see Appendix A.3 for a precise definition, and more details). To complete the proof of Theorem 4.1, we construct a linear mapping that maps the output of the last feedforward layer to a vector of logits that encodes the set of valid next words $S(\underline{w})$.

## 5.4 On the linear representation hypothesis

Our existence result is compatible with the empirical phenomenon known as the linear representation hypothesis (Park et al. (2024); Garg et al. (2026)), in the context of transformers that model context-free grammars. With experiments on synthetic data, the empirical study Allen-Zhu & Li (2025) shows that Transformers can learn the recursive rules of context-free grammars, with abstract non-terminal symbols being recoverable through linear probes applied to the embeddings in the final transformer layer. Using experiments on real-world datasets, the empirical study Saglam et al. (2025) shows that transformers encode high-level semantic representations within low-dimensional subspaces in hidden transformer layers.

Our explicit construction shows that transformer architectures possess the structural capacity to explicitly represent these complex grammatical structures within low-dimensional linear subspaces. For a fixed derivation tree shape $\underline{c} \in \mathcal{C}(\mathcal{G})$, the parsing computations are performed within low-dimensional subspaces in the

intermediate representation indexed by $\underline{c}$. Consequently, a simple affine transformation is mathematically sufficient to isolate and separate the representations of distinct non-terminal symbols. These theoretical observations about linear separability support the empirical findings in Allen-Zhu & Li (2025) about linear probes.

However, our construction encodes non-terminals in low-dimensional subspaces through a strictly orthogonal basis, allocating distinct indicator neurons for specific grammatical states. This simplified model does not account for the principle of superposition (Garg et al. (2026); Saglam et al. (2025)), where models compress more features than the available dimensions. It is an open question to understand how standard training dynamics yield transformers that compress grammatical structures into dense vectors in residual streams using superposition.

### 5.5 Example.

We revisit the CFG from Table 1, and provide two examples to illustrate the notation used in this section. Here $\underline{c}_1$ and $\underline{c}_2$ are the shapes corresponding to the two derivation trees in Figure 1.

$$\underline{w}_1 = (a_1, a_6, a_5, a_8, a_5, a_2, a_7)$$
$$E_{3,\underline{c}_1}(\underline{w}_1) = a_1^3, E_{2,\underline{c}_1}(\underline{w}_1) = a_3^2, E_{1,\underline{c}_1}(\underline{w}_1) = a_7$$
$$N_{3,\underline{c}_1}(\underline{w}_1) = \{a_4^3\}, N_{2,\underline{c}_1}(\underline{w}_1) = \{a_4^2\}$$
$$N_{1,\underline{c}_1}(\underline{w}_1) = \{a_4, a_8\}$$
$$\underline{w}_2 = (a_6, a_8, a_3, a_9, a_{10}, a_4, a_5,$$
$$a_3, a_1, a_7, a_9, a_7, a_8)$$
$$E_{3,\underline{c}_2}(\underline{w}_2) = a_2^3 a_3^3, E_{2,\underline{c}_2}(\underline{w}_2) = a_4^2, E_{1,\underline{c}_2}(\underline{w}_2) = \emptyset$$
$$N_{3,\underline{c}_2}(\underline{w}_2) = \{a_4^3\}, N_{2,\underline{c}_2}(\underline{w}_2) = \{a_5^2\}$$
$$N_{1,\underline{c}_2}(\underline{w}_2) = \{a_3, a_{10}\}$$

## 6 Discussion and limitations

**Interpretability of transformers: circuits.**

Complementing our theoretical results, Allen-Zhu & Li (2025) shows that transformer models can empirically learn CFGs when trained using stochastic gradient descent, and demonstrates that the transformer model's hidden states encode non-terminal symbols (see also Zhao et al. (2023)). While our results establish the existence of transformers with circuits that can model CFGs, their findings suggest that similar structures may emerge in practice through stochastic gradient descent, but without a detailed analysis. Bridging this gap remains a central challenge for future work, and could be approached using mechanistic interpretability. Algorithms such as ACDC (Conmy et al. (2023)) and EAP (Syed et al. (2023)) could be used to identify and analyze circuits (Olah et al. (2020); Elhage et al. (2021)), that are sparse computational subgraphs of models which capture their behavior for specific tasks.

**On the inductive bias of sparse attention mechanisms.** While our theoretical constructions rely on a sparse attention mechanism to efficiently capture the hierarchical structure of languages generated by CFGs of depth $d$, prior work has proposed a variety of sparse attention patterns designed to scale transformers to longer sequences (Roy et al. (2020); Beltagy et al. (2020); Yuan et al. (2025)). While these architectures have shown empirical success in tasks like document classification and question answering (Ainslie et al. (2020)), and local attention patterns have been used in vision transformers (Liu et al. (2021)), a rigorous understanding of their expressive power and inductive biases remains limited. The BigBird architecture (Zaheer et al. (2020)) introduces structured sparsity in attention, which is a mixture of global, local, and random patterns, with provable guarantees under certain conditions. In future work, it would be interesting to extend our theoretical framework and characterize which classes of CFGs can be modeled using different sparse attention patterns, building on the analysis in Yun et al. (2020).

**Generalizations to question-answering tasks with transformer models.** While our framework focuses on sentence completion using CFGs, many of the tasks at which large language models excel — such as question answering, text or code generation and machine translation — also exhibit rich hierarchical structure that could be captured using extensions of CFGs. For instance, synchronous CFGs provide a natural formalism for modeling machine translation (Chiang (2005)). Attribute grammars augment CFGs with semantic attributes and rules (Knuth (1968)), and can model the dependencies and constraints present in question-answering tasks (Marion et al. (2021)), such as resolving references or selecting the correct scope of a question. Text and code generation tasks also involve hierarchical structures (Yin & Neubig (2017); Sun et al. (2020)), and extending our theoretical constructions to these richer grammar classes represents a promising direction for understanding and improving the generalization capabilities of deep models across a wider range of language tasks.

## 7 Conclusion

This paper provides a theoretical framework for understanding the expressiveness of deep transformer-based language models by analyzing their ability to model hierarchical structures using a class of CFGs with fixed depth and a bounded number of tree shapes. We give an explicit construction of transformers with a positional attention mechanism, where the model depth increases linearly with the depth of the underlying CFG, while the number of neurons required in each layer scales quadratically with the number of production rules and linearly with the number of derivation tree shapes. These results formalize the intuition that deeper architectures can represent increasingly complex linguistic structures and mitigate the curse of dimensionality by exploiting hierarchical structures, mirroring the recursive decomposition of phrases and clauses observed in natural language.

Our expressivity results support recent empirical observations in Allen-Zhu & Li (2025); Saglam et al. (2025) about the linear representation hypothesis using an orthogonal basis space in a simplified model without superposition. We prove that transformers have the structural capacity to encode non-terminal symbols using low-dimensional subspaces in their hidden layers, and implement bottom-up dynamic programming algorithms for parsing CFGs, with each layer of the transformer corresponding to a level in the grammatical hierarchy. This work lays the groundwork for further theoretical investigation into the inner workings of transformer models and could lead to new insights into the design of more efficient, interpretable architectures informed by formal language theory.

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

# A  Proof of Theorem 4.1

## A.1  Preliminary lemmas

In this section, we recall results from Zhang et al. (2017) and Yun et al. (2019), which analyze the memorization capacity of feedforward neural networks, and show that they can fit an arbitrary finite set of input-output vectors. These results will be used in the proof of Lemmas 5.2 and A.6 below.

**Lemma A.1.** *Let $\{(x_i, y_i)\}_{i=1}^N$ be a dataset where all $x_i$ are distinct and each label satisfies $y_i \in [-1, 1]^{d_y}$. Suppose $f_\theta$ is a feedforward neural network with two hidden layers, ReLU activations, input dimension $d_1$, hidden width $d_2$, and output dimension $d_y$. If*

$$4 \left\lfloor \frac{d_1}{4} \right\rfloor \left\lfloor \frac{d_2}{4 d_y} \right\rfloor \geq N,$$

*then there exists a choice of parameters $\theta$ such that*

$$f_\theta(x_i) = y_i \quad \text{for all } i \in [N].$$

We will also need the following basic Lemma in the proofs for Section 5.3.

**Lemma A.2.** *Fix $k \in \mathbb{N}$ and choose distinct points $t_1, \ldots, t_k \in \mathbb{R}$. There exists $\delta > 0$ and a one-hidden-layer ReLU network $f : \mathbb{R} \to \mathbb{R}$ with $3k$ hidden units such that:*

- *$f(t_m) = 1$ for each $m \in \{1, 2, \ldots, k\}$,*

- *$f(x) = 0$ for all $x \notin \bigcup_{m=1}^k [t_m - \delta, t_m + \delta]$,*

- *$f$ is piecewise linear with exactly one triangular peak on each interval $[t_m - \delta, t_m + \delta]$.*

*Proof.* For each $m \in \{1, \ldots, k\}$ define the triangular "hat" function

$$h_m(x) = \frac{1}{\delta} \Big( \text{ReLU}\big(x - (t_m - \delta)\big) - 2 \, \text{ReLU}(x - t_m) + \text{ReLU}\big(x - (t_m + \delta)\big) \Big).$$

Each $h_m$ is supported on $[t_m - \delta, t_m + \delta]$, satisfies $h_m(t_m) = 1$, is linear on $[t_m - \delta, t_m]$ and $[t_m, t_m + \delta]$, and is 0 elsewhere. Set

$$f(x) = \sum_{m=1}^k h_m(x).$$

Choosing $\delta > 0$ small enough ensures the supports $[t_m - \delta, t_m + \delta]$ are pairwise disjoint, so at any $x$ at most one $h_m$ is nonzero. Hence $f(t_m) = h_m(t_m) = 1$ for each $m$, $f(x) = 0$ off $\bigcup_{m=1}^k [t_m - \delta, t_m + \delta]$, and $f$ is piecewise linear with one triangular peak on each such interval. The representation uses $3k$ ReLU units in a single hidden layer, completing the construction. $\qquad \square$

## A.2 Proof of Lemma 5.1 and Lemma 5.2

We start with a rigorous definition of the quantities $E_{i,\underline{c}}(\underline{w}), E'_{i,\underline{c}}(\underline{w})$ and $E^*_{i,\underline{c}}(\underline{w})$ introduced in Section 5.1.

**Definition A.3.** Let $G = (\Sigma, N, P, S)$ be a context–free grammar of depth $d$ and fix a derivation-tree shape $\underline{c}$. For an input sequence $\underline{w} = w_1 \cdots w_n \in \Sigma^*$ define a sequence of strings $R^{(1)}(\underline{w}), R^{(2)}(\underline{w}), \ldots, R^{(d)}(\underline{w}) \subseteq (\Sigma \cup N)^*$ inductively:

1. $R^{(1)}(\underline{w}) := \underline{w}$, which corresponds to the leaves of the tree at depth 1.

2. For each level $i = 1, \ldots, d - 1$, construct $R^{(i+1)}(\underline{w})$ from $R^{(i)}(\underline{w})$ as follows. Traverse the derivation tree shape $\underline{c}$ at level $i + 1$. Whenever a non-terminal $A$ at this level is the parent of a contiguous sequence of children $\gamma$ appearing in the derivation tree for $R^{(i)}(\underline{w})$, replace those children by their parent node $A$. If there is no such non-terminal symbol for that sequence of children, then we replace the sequence $\gamma$ with the null symbol $\emptyset$. This operation is well-defined because $\underline{c}$ fixes all parent–child relations and their left-to-right ordering in the tree. Hence the resulting string $R^{(i+1)}(\underline{w})$ is unique.

The resulting string $R^{(d)}(\underline{w})$ can be expressed uniquely as follows. Here $E_{i,\underline{c}}(\underline{w})$ is the sequence consisting of non-terminal symbols at level $i$, and possibly some null symbols, appearing in the string $R^{(d)}(\underline{w})$.

$$R^{(d)}(\underline{w}) = E_{d,\underline{c}}(\underline{w}) \cdots E_{i,\underline{c}}(\underline{w}) \cdots E_{1,\underline{c}}(\underline{w}).$$

If $E_{i,\underline{c}}(\underline{w})$ is not the empty string, define $E^*_{i+1,\underline{c}}(\underline{w}) = \emptyset$, and $E'_{i,\underline{c}}(\underline{w}) = E_{i,\underline{c}}(\underline{w})$. If $E_{i,\underline{c}}(\underline{w})$ is the empty string, we define $E'_{i,\underline{c}}(\underline{w})$ as the string obtained by concatenating all non-terminals in the $i$-th level of the derivation tree for $\underline{w}$ with the same parent as the last non-terminal symbol in that level, and define $E^*_{i+1,\underline{c}}(\underline{w})$ to be the corresponding parent symbol in the $i + 1$-st level of the derivation tree. ∎

**Definition A.4.** Given a class $\underline{c} \in \mathcal{C}(\mathcal{G})$, let $\mathcal{S}_i(\underline{c})$ be the set of all strings $E_{i,\underline{c}}(\underline{w})$ obtained from valid prefixes $\underline{w}$ corresponding to the class $\underline{c} \in \mathcal{C}(\mathcal{G})$.

**Example.** We revisit the context-free grammar from Table 1, and illustrate the above notation with an example. Here $\underline{c}_2$ is the shape corresponding to the second derivation tree in Figure 1.

$$\underline{w}_2 = (a_6, a_8, a_3, a_9, a_{10}, a_4, a_5, a_3, a_1, a_7, a_9, a_7, a_8)$$
$$R^{(1)}(\underline{w}_2) = a_3^2 a_4^2 a_5^2 a_1^2 a_2^2 a_4^2, R^{(2)}(\underline{w}_2) = a_2^3 a_3^3 a_4^2$$
$$E_{3,\underline{c}_2}(\underline{w}_2) = E'_{3,\underline{c}_2}(\underline{w}_2) = a_2^3 a_3^3, E^*_{3,\underline{c}_2}(\underline{w}_2) = \emptyset, E_{2,\underline{c}_2}(\underline{w}_2) = E'_{2,\underline{c}_2}(\underline{w}_2) = E^*_{2,\underline{c}_2}(\underline{w}_2) = a_4^2$$
$$E_{1,\underline{c}_2}(\underline{w}_2) = \emptyset, E'_{1,\underline{c}_2}(\underline{w}_2) = a_7 a_8 \quad ∎$$

We now prove Lemma 5.1 and Lemma 5.2 using a joint inductive argument.

*Proof of Lemma 5.1.* We proceed by induction on the layer, for a given input sequence $\underline{w} = w_1 \cdots w_n \in \Sigma^*$. For the induction step, we assume that the claim in Lemma 5.2 is true for the previous layer - i.e. that for each $j \leq i$, the $j$-th coordinate of $z_{i,\underline{c}}(\underline{w})$ depends only on the string $E_{j,\underline{c}}(\underline{w})$.

The relative positional weights $R_h$ for the attention head indexed by $\underline{c}$ are chosen so that the last token attends only to earlier tokens corresponding to the non-terminal symbols in $E'_{i+1,\underline{c}}(\underline{w})$. Because we omit the softmax function and instead use a positional attention mechanism, the other attention scores are zero.

We choose the value matrices $W_h^{(V)}$ so that the output $a^{\underline{c}}_{i+1}(\underline{w})$ of the attention head indexed by $\underline{c}$ only depends on the $i$-th coordinate of the vectors $z_{i,\underline{c}}(\underline{w}_{1:j})$ for $1 \leq j \leq n$. This can be done by having non-zero values in the column of the value matrix $W_h^{(V)}$ corresponding to that coordinate, and setting all other columns to zero. By the induction hypothesis, it now follows that the output $a^{\underline{c}}_{i+1}(\underline{w})$ depends only on the quantity $E'_{i+1,\underline{c}}(\underline{w})$. □

In order to prove a similar statement without using the simplified relative positional encoding, one can follow the approach from the proof of Theorem 4.1 in Svete & Cotterell (2024), and use query-key matrices that

encode positional information. This is achieved by placing the positional embeddings in a subspace that is disjoint from the word embeddings. The query–key matrices act only on the positional subspace, so that the dot product between the query vector and key vector depends only on positional information.

Given an input sequence $\underline{w} = (w_j)_{1 \le j \le k}$, we denote by $z_i'(\underline{w})$ its image after passing through the $W^{(O)}$ matrix in the $i$-th transformer block. We decompose these vectors $z_i'(\underline{w})$ into $c$ blocks as follows: $z_i'(\underline{w}) = \{z_{i,\underline{c}}'(\underline{w})\}_{\underline{c} \in \mathcal{C}(\mathcal{G})}$. Here $z_{i,\underline{c}}'(\underline{w}) \in \mathbb{R}^d$ for each $\underline{c} \in \mathcal{C}(\mathcal{G})$.

*Proof of Lemma 5.2.* We again proceed by induction on the layer $i$. For the induction step, we assume that the claim in Lemma 5.1 is true for the previous layer - i.e. that the output vector $a_i^{\underline{c}}(\underline{w})$ of the attention head indexed by $\underline{c}$ in the $i$-th layer depends only on $E_{i,\underline{c}}'(\underline{w})$.

We choose the concatenation matrix $W^{(O)}$, so that the following holds: for each $j < i$, the $j$-th coordinate of $z_{i,\underline{c}}'(\underline{w})$ depends only on the string $E_{j,\underline{c}}(\underline{w})$, the $i$-th coordinate of $z_{i,\underline{c}}'(\underline{w})$ depends only on the string $E_{i,\underline{c}}'(\underline{w})$, and the $j$-th coordinate of $z_{i,\underline{c}}'(\underline{w})$ is 0 when $j > i$. This is done when $j \neq i$ using the residual skip connection, whereby the corresponding coordinates of $z_{i-1,\underline{c}}'(\underline{w})$ already satisfy these constraints, and the $W^{(O)}$ matrix leaves those coordinates unchanged. When $j = i$, the $W^{(O)}$ matrix maps the vectors $a_i^{\underline{c}}(\underline{w})$ to the $i$-th coordinate of $z_{i,\underline{c}}'(\underline{w})$; from Lemma 5.1, this quantity depends only on the string $E_{i,\underline{c}}'(\underline{w})$.

To complete the proof, we use the position-wise feedforward network to modify the $i$-th and $i+1$-st coordinates of $z_{i,\underline{c}}'(\underline{w})$, and leave all other coordinates unchanged. We construct **FF** using Lemma 5.3 above, so that it satisfies the following for each $\underline{w}$ (note that the superscripts denote the coordinate index of the corresponding vector in $\mathbb{R}^d$). Here the $i$-th coordinate $z_{i,\underline{c}}^{(i)}(\underline{w})$ depends on the string $E_{i,\underline{c}}(\underline{w})$, while the $(i+1)$-st coordinate $z_{i+1,\underline{c}}^{(i)}(\underline{w})$ depends only on the string $E_{i+1,\underline{c}}^*(\underline{w})$, and is 0 if $E_{i+1,\underline{c}}^*(\underline{w}) = \emptyset$.

$$\mathbf{FF}(\cdots, z_{i,\underline{c}}^{'(i)}(\underline{w}), \cdots) = (\cdots, z_{i,\underline{c}}^{(i)}(\underline{w}), z_{i+1,\underline{c}}^{(i)}(\underline{w}), \cdots)$$

In Lemma 5.3, note that the feedforward network requires at most $M$ neurons, because there are at most $M$ different values of $z_{i,\underline{c}}^{'(i)}(\underline{w})$, corresponding to the possible values of $E_{i,\underline{c}}'(\underline{w})$. Note also that for each $j < i$, the $j$-th coordinate of $z_{i,\underline{c}}(\underline{w})$ depends only on the string $E_{j,\underline{c}}(\underline{w})$, and the $j$-th coordinate of $z_{i,\underline{c}}(\underline{w})$ is 0 when $j > i + 1$, as the residual skip connection ensures that these coordinates remain unchanged. $\square$

### A.3   Using stacked feedforward networks for next-word prediction: proofs

We elaborate on the notation introduced in Section 5.3, and define the sets $N_{d-i+1,\underline{c}}(\underline{w})$ rigorously below. In Lemma A.6 below, we show how these can be computed using a feedforward neural network.

**Definition A.5.** Given an input sequence $\underline{w}$ and $1 \le i \le d$, consider the string of symbols obtained by concatenating $E_{d,\underline{c}}(\underline{w})$, $E_{d-1,\underline{c}}(\underline{w})$, ..., $E_{d-i+1,\underline{c}}(\underline{w})$. We define the set $N_{d-i+1,\underline{c}}(\underline{w})$ as the set of all non-terminals of $N_{d-i+1}$ that can be appended to this string to yield a sequence which can be expanded into a valid sentence.

**Lemma A.6.** *We construct a feedforward neural network with $5d + 7$ hidden layers that has the following properties; here $h_i(\underline{w})$ denotes the image of an input sequence $\underline{w}$ after passing through the $i$-th feedforward layer in the sequence. Each of the layers has at most $4cBM^2$ neurons. Recall that $M$ is the maximal rule count, $B$ is the maximal branching factor, and $c = |\mathcal{C}(\mathcal{G})|$.*

- *The neurons of $h_{1,\underline{c}}$ are divided into $d$ blocks. For each $1 \le i \le d$, the $i$-th block of $h_{1,\underline{c}}$ has $|\mathcal{S}_i(\underline{c})|$ neurons.*

- *The block $h_{2,\underline{c}}$ of the second layer indexed by the class $\underline{c}$ has $d$ neurons which satisfy the following conditions. If $E_{j,\underline{c}}(\underline{w})$ does not contain any null symbols, then the value of the $j$-th neuron of $h_{2,\underline{c}}(\underline{w})$ is equal to the $j$-th coordinate of $z_{d,\underline{c}}(\underline{w})$. If $E_{j,\underline{c}}(\underline{w})$ contains any null symbols, then the value of the $j$-th neuron of $h_{2,\underline{c}}(\underline{w})$ is 0.*

- *For each $0 \le i \le d$, the neurons of $h_{5i+2,\underline{c}}(\underline{w})$ are divided into $d$ blocks. For each $j \le d - i$, the $j$-th block of $h_{5i+2,\underline{c}}(\underline{w})$ has a single neuron, whose value is equal to the $j$-th coordinate of $h_{2,\underline{c}}$. For each*

$j > d - i + 1$, the $j$-th block of $h_{5i+2,\underline{c}}(\underline{w})$ has a single neuron, which is equal to $0$. If $i > 0$, the $(d - i + 1)$-st block of $h_{5i+2,\underline{c}}$ has $|N_{d-i+1}|$ neurons. In this block, a neuron in $h_{5i+2,\underline{c}}(\underline{w})$ has value $1$ if the corresponding symbol in $N_{d-i+1}$ is in the set $N_{d-i+1,\underline{c}}(\underline{w})$, and $0$ otherwise.

- For each $0 \leq i \leq d - 1$, the neurons of $h_{5i+3,\underline{c}}(\underline{w})$ and $h_{5i+4,\underline{c}}(\underline{w})$ are divided into $d$ blocks. For each $j \neq d - i + 1$, the $j$-th blocks of $h_{5i+3,\underline{c}}(\underline{w})$ and $h_{5i+4,\underline{c}}(\underline{w})$ have a single neuron, which takes the same value as the corresponding neuron in $h_{5i+2,\underline{c}}(\underline{w})$. If $i > 0$, the $(d - i + 1)$-st blocks of $h_{5i+3,\underline{c}}(\underline{w})$ and $h_{5i+4,\underline{c}}(\underline{w})$ are divided into sub-blocks, with one sub-block for each non-terminal symbol in $N_{d-i+1}$. Given $n \in N_{d-i+1}$, the number of neurons in the corresponding sub-block of $h_{5i+3,\underline{c}}(\underline{w})$ (resp. $h_{5i+4,\underline{c}}(\underline{w})$) is equal to $4|\mathcal{S}_{d-i}(\underline{c})|$ (resp. $4|N_{d-i}|$).

- For each $0 \leq i \leq d-1$, the neurons of $h_{5i+5,\underline{c}}(\underline{w})$ are divided into $d$ blocks. For each $j \leq d-i-1$, the $j$-th block of $h_{5i+5,\underline{c}}(\underline{w})$ has a single neuron, whose value is equal to the $j$-th coordinate of $h_{2,\underline{c}}(\underline{w})$. For each $j \geq d - i + 1$, the $j$-th block of $h_{5i+5,\underline{c}}(\underline{w})$ has a single neuron, which is equal to $0$. The $(d - i)$-st block of $h_{5i+5,\underline{c}}(\underline{w})$ has $|N_{d-i}|$ neurons.

- For each $0 \leq i \leq d-1$, the neurons of $h_{5i+6,\underline{c}}(\underline{w})$ are divided into $d$ blocks. For each $j \leq d-i-1$, the $j$-th block of $h_{5i+6,\underline{c}}(\underline{w})$ has a single neuron, whose value is equal to the $j$-th coordinate of $h_{2,\underline{c}}(\underline{w})$. For each $j \geq d - i + 1$, the $j$-th block of $h_{5i+6,\underline{c}}(\underline{w})$ has a single neuron, which is equal to $0$. The $(d - i)$-st block of $h_{5i+6,\underline{c}}(\underline{w})$ is divided into sub-blocks, with one sub-block for each non-terminal symbol in $N_{d-i}$, and each sub-block having at most $3M$ neurons.

*Proof of Lemma A.6.* We proceed by induction on $i$. For the base case, observe that $N_{d,\underline{c}}(\underline{w})$ can be computed directly from $E_{d,\underline{c}}(\underline{w})$. This computation is performed by the first block of the feedforward neural network. For the induction step, note that $N_{d-i,\underline{c}}(\underline{w})$ can be determined from $N_{d-i+1,\underline{c}}(\underline{w})$ together with $E_{d-i,\underline{c}}(\underline{w})$. This computation is carried out by the four layers in the $i$-th block of the feedforward neural network.

**Step 1.** We first construct the weights in the first two layers as follows. For each $i$ with $1 \leq i \leq d$, the neurons in the $i$-th block of $h_{1,\underline{c}}$ are connected to the $i$-th neurons of the input vector $z_{d,\underline{c}}(\underline{w})$, and the $i$-th neuron of the next layer $h_{2,\underline{c}}$. We construct the weights connecting these neurons using Lemma A.2. Recall that the $i$-th coordinate of $z_{d,\underline{c}}(\underline{w})$ is determined by the string $E_{i,\underline{c}}(\underline{w})$ if $\underline{w}$ has type $\underline{c}$, and this string is an element of $\mathcal{S}_i(\underline{c})$. In Lemma A.2, we set $k = \mathcal{S}_i(\underline{c})$, and set $t_1, \cdots, t_k$ to be the values of $z_{d,\underline{c}}(\underline{w})$ corresponding to the strings in $\mathcal{S}_i(\underline{c})$. We set $\delta$ to be sufficiently small so that if the string $E_{i,\underline{c}}(\underline{w})$ has null values, the corresponding value of $z_{d,\underline{c}}(\underline{w})$ is mapped to $0$ since it falls outside of the intervals $[t_m - \delta, t_m + \delta]$.

**Step 2.** For a given $i$, we now construct the weights connecting the neurons in layers $5i + 2, 5i + 3, 5i + 4$ and $5i + 5$. Our construction enforces this when $j \neq d - i, d - i + 1$ using an identity connection, as the value remains unchanged. For each non-terminal $n \in N_{d-i+1}$, consider the two neurons in $h_{5i+2,\underline{c}}(\underline{w})$ corresponding to $n$ and the $(d - i)$-th coordinate of $z_{d,\underline{c}}(\underline{w})$, the neurons in the sub-block of the $(d - i + 1)$-st blocks of $h_{5i+3,\underline{c}}(\underline{w})$ and $h_{5i+4,\underline{c}}(\underline{w})$ corresponding to $n$, and the neurons in the $(d - i)$-th block of $h_{5i+5,\underline{c}}(\underline{w})$. We choose the weights connecting these neurons using Lemma A.1 as follows.

For each non-terminal $n \in N_{d-i+1}$, we use Lemma A.1 with the input-output pairs $(x_s, y_s)_{i=1}^K$ chosen as follows. Here the input vectors are $x_s \in \mathbb{R}^2$, with the first coordinate being equal to the $(d-i)$-th coordinate of $z_{d,\underline{c}}(\underline{w})$ and the second coordinate being equal to either $1$ or $0$. The output vectors are $y_s \in \mathbb{R}^{|N_{d-i}|}$, where the coordinate corresponding to $n_{d-i} \in N_{d-i}$ takes the value $1$ if the second coordinate of $x_s$ is $1$ and there is a production rule for $n$ where the right-hand side starts with the string $E_{d-i,\underline{c}}(\underline{w}) + n_{d-i}$; and otherwise that coordinate takes the value $0$. Since the $(d - i)$-th coordinate of $z_{d,\underline{c}}(\underline{w})$ depends only on $E_{d-i,\underline{c}}(\underline{w})$, the total number of input-output pairs is equal to $2|\mathcal{S}_{d-i}(\underline{c})|$. Using Lemma A.1, we can memorize these input-output pairs using a feedforward network with two hidden layers where the first hidden layer has $4|\mathcal{S}_{d-i}(\underline{c})|$ neurons, and the second hidden layer has $4|N_{d-i}|$ neurons.

After using this construction to choose the weights connecting these neurons for each non-terminal $n \in N_{d-i+1}$, it follows by summing over these non-terminals that the neuron in $h_{5i+5,\underline{c}}(\underline{w})$ corresponding to a non-terminal symbol $n_{d-i} \in N_{d-i}$ takes the value $r$, where $r$ is the number of non-terminals in $N_{d-i+1,\underline{c}}(\underline{w})$ which have a production rule where the right-hand side starts with the string $E_{d-i,\underline{c}}(\underline{w}) + n_{d-i}$. The total number of neurons in $h_{5i+4,\underline{c}}(\underline{w})$ is $4|N_{d-i}||N_{d-i+1}|$, and the total number of neurons in $h_{5i+3,\underline{c}}(\underline{w})$ is $4|N_{d-i+1}||\mathcal{S}_{d-i}(\underline{c})|$.

Note that these quantities are less than $4BM^2$ (recall here that $B$ is the maximal branching factor, and $M$ is the maximal rule count for a fixed depth), since $|N_{d-i+1}| < M$ and $|\mathcal{S}_{d-i}(\underline{c})| < BM$.

**Step 3.** Recall that the $(d-i)$-th blocks of $h_{5i+5,\underline{c}}(\underline{w})$ and $h_{5i+7,\underline{c}}(\underline{w})$ have $|N_{d-i}|$ neurons. We choose the weights connecting the $(d-i)$-th blocks of $h_{5i+5,\underline{c}}(\underline{w})$ and $h_{5i+6,\underline{c}}(\underline{w})$, and the weights connecting the $(d-i)$-st blocks of $h_{5i+6,\underline{c}}(\underline{w})$ and $h_{5i+7,\underline{c}}(\underline{w})$, using Lemma A.2. This is done so that for each $n_{d-i} \in N_{d-i}$, if the corresponding neuron in $h_{5i+5,\underline{c}}(\underline{w})$ has value 0 (respectively, an integer value between 1 and $M$), then the corresponding neuron in $h_{5i+7,\underline{c}}(\underline{w})$ has value 0 (respectively, 1). We use Lemma A.2, with $t_j = j$ for each $1 \le j \le M$, to construct a block of $3M$ neurons in the layer $h_{5i+6,\underline{c}}(\underline{w})$, for each non-terminal $n_{d-i} \in N_{d-i}$, so that this condition is satisfied. $\qquad\square$

### A.4 Completing the proof of Theorem 4.1.

To complete the proof of Theorem 4.1, we construct a linear mapping sending the output of the final layer in the feedforward network constructed above to the vector of logits that encodes the next-word probabilities. Recall that the output of the final layer is $h_{5d+2}(\underline{w}) = (h_{5d+2,\underline{c}}(\underline{w}))_{\underline{c} \in \mathcal{C}(\mathcal{G})}$, where each $h_{5d+2,\underline{c}}(\underline{w})$ consists of $d$ blocks. For each $i \ge 2$, the $i$-th block has a single neuron which is 0. By construction, the first block has $|N_1|$ neurons, which correspond to the terminal symbols; a neuron has value 1 if the corresponding terminal symbol is in the set $N_{1,\underline{c}}(\underline{w})$ (and 0 otherwise). We construct a linear mapping that connects the neuron $h_{5d+2,\underline{c}}(\underline{w})$ corresponding to a given terminal symbol to the corresponding neuron in the output layer for each $\underline{c}$ with a fixed weight $\alpha$. For all other pairs of neurons, the weight has value 0.

It follows that the logits in the output layer corresponding to a given terminal symbol take a strictly positive value if the terminal symbol lies in $S(\underline{w})$, and are otherwise zero. Following Definition 3.4, for a given $\epsilon > 0$, we can choose a sufficiently large value of $\alpha$ so that the below inequality holds after applying the softmax. It follows that the transformer language model $f_{\mathcal{T}}$ predictively models the language $\mathcal{L}$ with error at most $\epsilon$, as required.

$$\sum_{w \notin S(\underline{w})} \hat{f}_{\mathcal{T}}(w \mid \underline{w}) \; < \; \epsilon.$$

