# OpenReview forum: "How can deep transformers represent hierarchical languages? An expressivity analysis via bounded-depth grammars."
_TMLR — Withdrawn by Authors_

### Review · Reviewer_Fwfw · 2026-05-21

**Summary Of Contributions:**

This paper studies the expressive power of deep transformer-based language models when the data-generating distribution has an explicit hierarchical structure. The authors model the hierarchical structure with bounded-depth, non-recursive context-free grammars, and formulate the task in terms of approximate next-word prediction. In this setup, a language model predictively models the language if it assigns at most $\epsilon$ total probability mass to invalid continuations for every valid prefix. This approximation-theoretic framing with bounded-depth languages is a useful perspective relative to much of the formal-language literature, which often emphasises exact recognition rather than probabilistic prediction, and directly tackles problems with unbounded depth.

The main contribution is a transformer construction whose depth scales linearly with the grammar depth, while the neuron count scales linearly with the number of tree shapes and quadratically with the number of production rules. The construction uses positional attention to aggregate relevant child constituents and feedforward layers to implement a bottom-up parsing procedure. This result formalises a transformer-to-language correspondence: successive transformer layers can represent progressively higher levels of a derivation tree, rather than memorising full strings.

A strength of the paper is that it connects several lines of work: expressivity analyses of transformers over formal languages, approximation advantages of deep (over shallow) networks, and empirical observations that transformers may encode syntactic or semantic variables in linearly accessible subspaces.

A key limitation is the particular notion of approximation used in the paper. The definition only controls the total probability mass assigned to invalid next-token continuations, and therefore does not require the model to assign the correct relative probabilities among valid continuations. As a result, the theorem establishes that the constructed transformer can avoid invalid continuations, but it does not show that it approximates the true conditional next-word distribution induced by the grammar.

**Additional Comments:**

None

**Audience:**

Yes

**Audience Explanation:**

There is a growing community using tools from formal language theory to better understand the capabilities and limitations of neural networks, especially transformer-based language models. This paper contributes to that line of work by giving a concrete expressivity result for deep transformers on a structured class of hierarchical languages.

The paper should also be relevant to readers interested in why depth helps in neural architectures and how transformers can represent compositional structure. While the setting is simplified, the result provides a useful theoretical model for reasoning about hierarchical structure in language modeling, and therefore fits well within TMLR’s broader interest in principled analyses of machine learning methods.

**Claims And Evidence:**

No

**Claims Explanation:**

Overall, the central technical claim seems to be supported once the formal setup is introduced, in particular the definitions in Section 3.1. The proof strategy is also coherent. However, I think the paper should be more explicit from the beginning about the particular notion of approximation being used. In Definition 3.4, approximation is defined only in terms of the total probability mass assigned to invalid continuations, which I expect to be substantially weaker than approximating the true next-token distribution induced by the grammar. Thus, the theorem shows that the constructed transformer can avoid invalid continuations up to $\epsilon$, but not necessarily that it approximates the true conditional word probabilities over the valid set. Since the paper motivates itself as taking an approximation-theoretic perspective on language modelling, this distinction should be made explicit in the introduction and statement of contributions, not only after the formal definition is introduced.

There are also several places where broader interpretive claims need additional support or clarification.

1. Near the end of Section 3.2, the statement that “natural language rarely exhibits deep recursive nesting due to limitations on human working memory” is intuitively plausible, but unsupported. Are there any references making quantitative statements on the typical depth of recursive structure? If not, a more cautious formulation is required.

2. The remark after Theorem 4.1 that "due to the dependency on $c$, our key result does not yield tight bounds for complex real-world corpora like the Penn Treebank" sounds vague. Does it mean that the number of distinct derivations $c$ in real corpora is much larger than the hidden dimension of realistic transformers?

3. The comparison with Svete & Cotterell’s result in Section 4.2 needs to be more self-contained. Since both papers apparently have a “Theorem 4.1,” referring to the prior result by theorem number is confusing; it would be clearer to refer to it as “Svete & Cotterell’s result” or something similar. More importantly, the claimed relationship between depth-2 CFGs and $n$-gram models needs more explanation. The paper introduces grammar depth, branching factor, rule count, and tree shapes, but it is not clear what “width $(k,1)$” means in this context, nor how exactly the correspondence between such CFGs and $n$-gram models should be formalised. The authors should define this notion of width and spell out the mapping.

4. The following paragraph comparing the CFG construction to finite-state automata also needs additional detail. The claim that converting these bounded-depth CFGs to an equivalent FSA requires a state space of size $O(|N|^d)$ is plausible, but the argument is only sketched. Since this comparison is used to justify why the present theorem does not follow directly from existing transformer/FSA expressivity results, it should be explained carefully. Additionally, the author's construction hides another potential $d$-dependence in the number of trees $c$: is this number expected to depend on the maximal tree depth? If yes, how? Should it be compared with the exponential growth of the number of states in the equivalent FSA?

**Requested Changes:**

The following changes, together with the address of the remarks raised in the claims section, are critical for securing my recommendation for acceptance.

1. **Add missing references on the expressivity advantage of depth.**
   The related work should include Ronen Eldan and Ohad Shamir, *The power of depth for feedforward neural networks*, COLT 2016, when discussing the expressivity advantage of deep over shallow networks.

2. **Add and discuss the related line of work on uniform-depth, non-recursive CFGs.**
   The paper should cite and compare against another line of work analysing neural networks through uniform-depth, non-recursive CFGs, starting with the PRX paper at https://journals.aps.org/prx/abstract/10.1103/PhysRevX.14.031001. This work already studies how deep networks trained with gradient descent can uncover latent hierarchical structure in classification tasks, quantitatively and before Allen-Zhu & Li. The discussion should also include later extensions to language modelling tasks, including https://proceedings.neurips.cc/paper_files/paper/2024/file/9740da1c07c7b451af14e11523f95271-Paper-Conference.pdf and https://icml.cc/virtual/2025/poster/45566, as well as the extension to grammars with multiple derivation trees in https://arxiv.org/abs/2602.06065. This line of work is particularly relevant and it is closer to the present setup than Allen-Zhu & Li in some respects, since it considers ensembles of grammars with fixed maximal rule count and maximal branching factor.

3. **Provide a more precise comparison to Liu et al.**
   The comparison with Liu et al. should be expanded and made more precise. In particular, the claims about the relationship between CFG depth and the number of states required by an equivalent finite-state automaton should be explained in detail, rather than only sketched.

4. **Revise Definition 3.9 to avoid notation conflict and clarify the attention mechanism.**
   In Definition 3.9, the symbol $d$ is used for the input dimension, but $d$ is also used throughout the paper for grammar depth. The authors should use a different symbol for the embedding dimension. In addition, the term “relative positional weights” is confusing, since the matrix entries appear to depend on absolute positions rather than relative distances. The authors should either clarify why the mechanism is relative or rename it accordingly.

5. **Use consistent terminology for the transformer embedding dimension.**
   The embedding dimension is sometimes referred to as “width” and sometimes as “hidden dimension,” including in Theorem 4.1. The authors should choose one term and use it consistently throughout the paper.

6. **Broaden the comparison after Theorem 4.1.**
   The paragraph after Theorem 4.1 compares the result with that of Zhao et al., emphasising that Zhao et al.’s construction has depth growing linearly with sentence length. The authors should also discuss Yao et al., since, although that work considers a simpler parenthesis-language setting, it also obtains depth linear in the depth of the language.

7. **Expanding comparison with empirical work.**
   The related line of work starting from https://journals.aps.org/prx/abstract/10.1103/PhysRevX.14.031001, and in particular https://journals.aps.org/pre/abstract/10.1103/qtd6-nl8p, also analyses the activations in the hidden layers of transformers and other architectures trained on uniform-depth grammars. This should be discussed in relation to the paper’s claims about linear representations and hidden-layer encodings of grammar structure.

8. **Add an explicit Appendix reference at the end of the first paragraph of Section 5.1.**
   The first paragraph of Section 5.1 introduces quantities whose rigorous definitions are deferred. It would help the reader to include an explicit link or reference to the relevant Appendix section at the end of that paragraph.

9. **Move or repeat the example in Section 5.5 closer to Figure 1.**
   The example in Section 5.5 is very useful, but it would be easier to understand if it appeared closer to Figure 1, or if a shorter version were included near the figure with a forward reference to the full example.

---

### Review · Reviewer_hxw7 · 2026-06-19

**Summary Of Contributions:**

This paper studies the expressivity of deep transformer language models for representing hierarchical languages. Specifically, the authors consider length-uniform, non-recursive context-free grammars with bounded depth, and provide an explicit transformer construction whose depth scales linearly with the depth of the grammar. The main theorem states that such a transformer, using a simplified positional attention mechanism, can predictively model the language generated by the grammar with arbitrarily small error. The construction is intended to formalize the intuition that deeper transformer layers can implement bottom-up hierarchical parsing, and to connect this view with recent empirical observations on linear representations in transformer residual streams.

The topic is timely and relevant to TMLR. I appreciate the attempt to provide a rigorous theoretical framework for hierarchical representations in transformers, and the paper is generally well motivated by recent empirical work on CFG learning and linear probes. The comparison with n-gram models and finite-state automata is also useful, as it helps position the result relative to simpler sequential models.

**Audience:**

Yes

**Audience Explanation:**

Yes. The paper would likely be of interest to researchers studying transformer expressivity, formal language theory, mechanistic interpretability, and theoretical explanations of hierarchical representations in language models. The question of how transformer depth relates to hierarchical grammar depth is important, and the paper provides a constructive result that may be useful as a stepping stone for future work.

In particular, the comparison with n-gram models and finite-state automata is potentially valuable. The paper argues that although the bounded-depth CFGs considered here can in principle be represented by finite-state automata, flattening the hierarchy into automaton states can lead to exponential growth, whereas the transformer construction uses depth to process grammatical reductions level by level. This is a meaningful perspective for understanding why layered architectures may be better suited to hierarchical structure than flat sequence models.

That said, the audience interest is somewhat limited by the restrictive assumptions and the gap between the constructed architecture and standard trained transformers. The paper is more likely to interest theoretically oriented readers than the broader TMLR audience looking for results directly explaining practical LLM behavior.

**Broader Impact Concerns:**

I do not see major ethical or societal risks arising directly from this work. The paper is theoretical and focuses on the expressivity of transformer architectures for formal languages. It does not introduce a deployed system, dataset involving human subjects, or application with immediate safety or fairness concerns.

**Claims And Evidence:**

No

**Claims Explanation:**

The paper provides a detailed constructive proof, and I believe the high-level direction is plausible. However, I do not find the current evidence fully convincing for the broader claims made in the paper.

First, the main theorem applies to a very restricted class of grammars: fixed-depth, uniform, non-recursive CFGs with a bounded number of derivation-tree shapes. While the authors acknowledge this restriction, the introduction and conclusion sometimes suggest a broader explanation of how deep transformers represent hierarchical language structure. Since the construction excludes recursive CFGs and depends explicitly on the number of tree shapes, the result does not directly address many difficult aspects of natural language hierarchy.

Second, the transformer model used in the construction is not a standard transformer in the usual sense. The attention mechanism is based on relative positional matrices whose weights depend only on position and can be hard-coded to select the desired syntactic constituents. This makes the proof closer to constructing a specialized parsing circuit than showing that ordinary learned self-attention naturally has the claimed representational capacity. The paper discusses how one might emulate this with standard attention, but this part remains informal and is not integrated into the main theorem.

Third, the link to the linear representation hypothesis is not fully supported. The constructed model uses orthogonal basis dimensions and essentially indicator-like encodings for grammatical states. This does show linear separability in a constructed representation, but it does not explain the dense, compressed, or superposed representations observed in empirical transformer models. The paper itself notes this gap, but the abstract and conclusion still present the result as supporting the linear representation hypothesis rather strongly.

Fourth, some parts of the proof rely on FFN memorization over finite sets of production-rule configurations. While this is mathematically valid for an existence proof, it weakens the claim that the construction captures a meaningful hierarchical inductive bias of transformers. More discussion is needed to separate what is achieved by hierarchical computation across layers from what is merely memorized by sufficiently wide feedforward networks.

Overall, the claims are partially supported at the level of a formal construction, but I do not think the evidence currently justifies the broader interpretation given in the paper.

**Requested Changes:**

1. The paper should more explicitly state that the result is an existence proof for a restricted class of bounded-depth, non-recursive, uniform CFGs under a simplified positional attention mechanism. Claims about natural language, standard transformers, and the linear representation hypothesis should be softened unless they are directly supported by the theorem.

2.  The main theorem currently relies on a positional attention mechanism with hard-coded relative positional matrices. The discussion of how to simulate this with standard softmax attention and positional embeddings should be made formal, or at least stated as a separate proposition with precise scaling in width, heads, and context length. Without this, the relevance to ordinary transformer architectures remains limited.

3. The assumptions of fixed depth, uniform rules, non-recursion, and bounded tree shapes are central to the proof. The paper should explain more clearly which assumptions are technically necessary, which are made for convenience, and whether any can be relaxed. In particular, the dependence on the number of derivation-tree shapes c should be discussed more carefully, since this quantity may be large for realistic grammars.

4. The proof uses FFNs as memorization devices for finite rule configurations. The authors should clarify how much of the expressive advantage comes from transformer depth and attention-based aggregation, versus finite lookup-table memorization in the FFN. A more explicit comparison with a purely feedforward or memorization-based construction would help.

5. Although this is a theoretical paper, a small synthetic experiment showing that a transformer with comparable architectural constraints can learn or emulate the constructed parser would make the contribution more concrete. This is not strictly necessary, but it would help bridge the gap between the construction and empirical transformer behavior.

---

### Review · Reviewer_ExwY · 2026-07-05

**Summary Of Contributions:**

This paper studies the expressivity of deep transformers for representing hierarchical languages generated by bounded-depth, non-recursive context-free grammars. The main contribution is an explicit construction of a decoder-style transformer with positional attention whose depth scales linearly with the grammar depth. The construction implements a bottom-up parsing procedure: attention heads aggregate local syntactic constituents according to derivation-tree shapes, while feedforward layers act as lookup tables for production rules and encode non-terminal symbols in the residual stream.

A strength of the paper is that it provides a clean theoretical framework connecting transformer depth, hierarchical compositionality, and the linear representation hypothesis. The comparison with n-grams, finite-state automata, and prior CFG-parsing results is also useful. However, the result is limited by several strong assumptions: the grammar is bounded-depth, non-recursive, and uniform; the attention mechanism is simplified and largely hard-coded; and the construction establishes existence rather than learnability by gradient descent.

**Audience:**

Yes

**Audience Explanation:**

The paper should be of interest to researchers working on theoretical understanding of transformers, formal language expressivity, mechanistic interpretability, and hierarchical representation learning. The result provides a useful bridge between empirical observations about CFG learning in transformers and formal expressivity analysis.

**Claims And Evidence:**

Yes

**Claims Explanation:**

The paper clearly states a formal theorem and provides a constructive proof strategy showing how transformer layers can simulate bottom-up parsing for the specified class of CFGs. The construction is plausible and aligns with the stated assumptions.

That said, some claims would benefit from more precise qualification. In particular, the connection to the linear representation hypothesis should be presented as a compatibility result rather than an explanation of how trained transformers actually form such representations. The construction uses orthogonal, explicitly allocated features, whereas empirical transformer representations are usually dense and may involve superposition. Similarly, the discussion of natural language should be softened, since bounded-depth non-recursive CFGs are only a simplified abstraction of natural language syntax.

**Requested Changes:**

1. The authors should emphasize more explicitly that the result applies to bounded-depth, non-recursive, uniform CFGs, and that it does not directly address recursive CFGs or full natural language syntax.

2. Soften the claims related to the linear representation hypothesis. The paper should make clear that the construction supports the structural possibility of linear representations, but does not explain why SGD-trained transformers would learn them or how dense superposition emerges.

3. Clarify the role of simplified positional attention. The paper should better distinguish what is proved for the simplified attention mechanism and what would be required to extend the result to standard softmax attention.

4. Improve readability of the proof outline. A small worked example tracing how one prefix is processed across transformer layers would make the construction easier to follow.

---

### Note · Authors · 2026-07-16

**Comment:**

Dear TMLR reviewers,

We are grateful for all your feedback, and will resubmit our manuscript to TMLR soon after the feedback has been addressed.

Thanks.

**Withdrawal Confirmation:**

I have read and agree with the venue's withdrawal policy on behalf of myself and my co-authors.